# Classification of Subgroups with Immune Characteristics Based on DNA Methylation in Luminal Breast Cancer

**DOI:** 10.3390/ijms232112747

**Published:** 2022-10-22

**Authors:** Mengyan Zhang, Te Ma, Cong Wang, Jiyun Zhao, Jie Xing, Honghao Liu, Mu Su, Ruiyang Zhai, Ting Liu, Baoqing Sun, Yan Zhang

**Affiliations:** 1School of Life Science and Technology, Computational Biology Research Center, Harbin Institute of Technology, Harbin 150001, China; 2College of Pathology, Qiqihar Medical University, Qiqihar 161042, China; 3State Key Laboratory of Respiratory Disease, Guangzhou Medical University, Guangzhou 510089, China

**Keywords:** luminal breast cancer, subgroup, DNA methylation, immune microenvironment, scRNA-seq, pathology features

## Abstract

Luminal breast cancer (BC) accounts for a large proportion of patients in BC, with high heterogeneity. Determining the precise subtype and optimal selection of treatment options for luminal BC is a challenge. In this study, we proposed an MSBR framework that integrate DNA methylation profiles and transcriptomes to identify immune subgroups of luminal BC. MSBR was implemented both on a key module scoring algorithm and “Boruta” feature selection method by DNA methylation. Luminal A was divided into two subgroups and luminal B was divided into three subgroups using the MSBR. Furthermore, these subgroups were defined as different immune subgroups in luminal A and B respectively. The subgroups showed significant differences in DNA methylation levels, immune microenvironment (immune cell infiltration, immune checkpoint PD1/PD-L1 expression, immune cell cracking activity (CYT)) and pathology features (texture, eccentricity, intensity and tumor-infiltrating lymphocytes (TILs)). The results also showed that there is a subgroup in both luminal A and B that has the benefit from immunotherapy. This study proposed a classification of luminal BC from the perspective of epigenetics and immune characteristics, which provided individualized treatment decisions.

## 1. Introduction

BC is a heterogeneous disease that affects human health. Precise subgrouping of BC is beneficial for its treatment. According to the molecular subtype method, BC can be divided into five subtypes: luminal epithelial A (luminal A), luminal epithelial B (luminal B), HER2 overexpression, basal-like, and normal-like subtypes [1]. Indeed, luminal BC, including luminal A and luminal B, is more common in BC patients, accounting for 2/3 of all BC [2,3] and has significant heterogeneity [4,5,6]. The best treatment regimen for patients with luminal A/B-subtype tumors remains undetermined owing to the molecular and clinical heterogeneity [7,8]. Therefore, it is necessary to explore the differences in the internal luminal A/B subtype deeply.

Recently, more research has been conducted on the BC subtypes. For example, Curtis et al. identified ten different subtypes of BC [9]. Kroemer et al. designed different treatment plans for patients with luminal A and raised the question of whether it is necessary to reduce the level of treatment and omit chemotherapy in some cases [10]. Netanely et al. used RNA-Seq and DNA methylation data to perform unsupervised clustering of 1148 and 679 BC samples into different prognostic subgroups [11]. In 2011, Bediaga, et al. used an unsupervised clustering method to separate the whole luminal A sample from luminal B-HER2, and luminal B tumors were divided into two groups, which were grouped together with luminal A and luminal B-HER2 [12]. These studies showed that there were different subgroups within the BC subtypes, and these subgroups were different in terms of prognosis and treatment.

Epigenetic transcriptional regulation of DNA methylation is closely related to the corresponding gene expression in the human genome. DNA methylation profiles have been shown to be disturbed in cancer [13,14,15,16], and DNA methylation analysis has been used to classify many human malignancies and monitor cancer progression based on tumor-specific methylation signals [17]. Stirzaker et al. classified TNBC patients into one of three categories based on their disease results (bad, moderate, and good), thereby determining the DNA methylation characteristics of TNBC patients [18]. Therefore, DNA methylation is an important biomarker for accurate subtyping of BC.

In this study, we developed a computational framework MSBR, for the luminal A/B subgroup. Luminal A was divided into immune-cell enriched (AIE)/epithelial cells immune-cell enriched (AEE) subgroup, and luminal B was divided into immune-cell enriched (BIE)/epithelial cells immune-cell enriched (BEE)/None-cell enriched (BNE) subgroups based on DNA methylation, and AIE/BIE was predicted to have a better immune response. This study explained the heterogeneity of the luminal BC immune microenvironment and illustrated that the MSBR framework provided clinical decisions for the diagnosis and treatment of luminal A/B.

## 2. Results

### 2.1. Overview of the Study

This study comprehensively investigated the molecular and imaging characteristics of luminal BC subgroups and identified subgroups that benefit from immunotherapy. First, we analyzed HM450K DNA methylation data downloaded from the TCGA database. The heatmaps showed different DNA methylation levels in cancer vs. normal, luminal vs. other BC subtypes, and luminal A vs. luminal B. The results reflected the features of DNA methylation patterns in luminal A or luminal B. (Appendix A). Therefore, we developed an MSBR framework for subgrouping luminal A and luminal B subtypes (Figure 1). The *T* test was used to obtain the differentially methylated DNA genes of luminal A/B and normal samples for determining co-methylation modules by WGCNA. A function-based calculation method was used to screen the key modules in luminal A/B from the above modules. Gene sets from the key modules were obtained. We then divided the luminal A/B samples in a 7:3 ratio between the training set and the validation set. In the training set, the differentially methylated DNA sites were mined by the *t*-test. The “Boruta” algorithm was used for feature selection of differentially methylated DNA sites to construct a support vector machine classifier (SVM) for luminal A/B, which produced a feature gene set. Finally, the intersection of the feature gene set and key module gene set was used as the “key genes” for dividing into subgroups of luminal A/B based on hierarchical clustering. The scRNA-seq data were used to perform deconvolution calculation to define subgroups (see methods) (Figure 1A). We also analyzed the differences of in survival, clinical stage, immune microenvironment, and pathological image features between the luminal A/B subgroups (Figure 1B).

### 2.2. Acquisition of “Key Genes” from MSBR Framework

According to the MSBR framework, we first received 7668/12,717 differential DNA methylation sites (2183/4637 differential DNA methylation genes) obtained from all 349/150 luminal A/B samples and 96 normal samples. The differential DNA methylation genes were significantly enriched in the Wnt signaling pathway, Ras signaling pathway, and breast cancer (Appendix A). The 2183/4637 differential DNA methylation genes were analyzed using the WGCNA module, and a total of five and six modules were obtained in luminal A and luminal B, respectively, including 251/512 genes in the brown module, 50/160 genes in the yellow module, 406/1505 genes in the turquoise module, 291/770 genes in the blue module, and 91 genes in the green module for luminal B. The grey module could not be clearly classified into any module (Figure 2A,C). We performed pathway enrichment analysis and computed the important scores KMscore of each module according to Algorithm 1. By calculating the ranking, the two blue modules were defined as the key functional modules in luminal A and luminal B.
**Algorithm 1**: Selection of key genes for subgrouping. **Input:** KEGG pathways results (kegg pathway kegg, *p*-value P, genes sets Gene, module number N, gene sets ${\mathrm{Gene}}_{i}$ in module i, feature DNA methylation genes $Genes_{feature}$. **Output:** key genes KG1 Take all the kegg pathways with *p* < 0.05 as set KP2 Obtain all genes in set KP as KPGenes3 **for**i=1 to N **do**4  Count the intersection of KPGenes and $Gene_{i}$, then divided by the number of elements of set $Gene_{i}$ as $KMscore_{i}$.5 **end**6 Define an empty set KMGenes7 **for**i=1 to N **do**8  **if**$KMscore_{i}$ > the mean of {$KMscore_{1}$, $KMscore_{2}$, …, $KMscore_{N}$} **and**
$KMscore_{i}$ ≥ *upper quartile of set* {$KMscore_{1}$, $KMscore_{2}$, …, $KMscore_{N}$} **then**9    Add set $Gene_{i}$ into set KMGenes10   **end**
11 **end**12 Acquire set KG as the intersection of KMGenes and $Genes_{feature}$

We then divided 349/150 luminal A/B samples and 343/542 non-luminal A/B into the training set and the validation set according to 7:3 and obtained 7776/12,166 differential DNA methylation sites by performing differential DNA methylation analysis between luminal A/B and normal samples. Then, the “Boruta” method was used to obtain 200/258 differential DNA methylation sites for luminal A/B and normal samples. Finally, 200/258 differential DNA methylation sites were screened out as the main features of the SVM classifiers in luminal A/B and non-luminal A/B. Ten-fold cross-validation was applied for training and validation. The AUCs of the training set, test set, and validation set in the luminal A/B classifiers were 0.921/0.904, 0.820/0.864, and 0.814/0.739, respectively (Appendix A). Finally, by taking the intersection with the feature DNA methylation genes of the classifier, 22/11 key DNA methylation genes were obtained and were used as the “key genes” for subgrouping luminal A/B (Figure 2B,D).

### 2.3. The Definition of Subgroups in Luminal A/B

A hierarchical clustering method was used to identify subgroups based on the DNA methylation levels of the 22/11 “key genes”. Finally, two/three subgroups were defined as luminal A/B according to the MSBR framework (Appendix A). The TSNE method displayed luminal A/B subgroups, and it could be seen that the two and three subgroups were well separated (Figure 3A). In addition, we used five cases of BC single-cell sequencing data from GSE180286 [19] to definite characteristics of subgroups, and “Seurat” and “SingleR” were applied to obtain 17 clusters and finally annotate seven types of cell types (B cells, Chondrocytes, Endothelial cells, Epithelial cells, Macrophage, T cells and tissue stem cells) in BC. By calculation, we obtained a signature matrix of seven cell types and fed the signature matrix into the deconvolution algorithm to obtain the cell proportions for the seven cell types (See Methods). In luminal A, the proportion of T cells and tissue stem cells was significantly higher in cluster1, and the proportion of epithelial cells was significantly higher in cluster2; in luminal B, the proportion of epithelial cells was significantly higher in cluster3, and the proportion of T cells was significantly higher in cluster1 (Appendix A). Therefore, two subgroups of luminal A were defined as immune-cell enriched (AIE)/epithelial cells immune-cell enriched (AEE) and three subgroups in luminal B were defined as immune-cell enriched (BIE)/epithelial cells immune-cell enriched (BEE)/None-cell enriched (BNE) subgroups by 22 and 11 key genes. AIE and AEE were significantly different in survival. BIE, BEE, and BNE also showed significant differences in survival (*p* < 0.05) (Figure 3B,C). The counting percentage of the clinical stages (T/N/M) of each subgroup showed that BEE tended to have more T4 and T3 samples and AIE had more T1 samples (Figure 3D). To demonstrate that the subgroups obtained by the MSBR framework were meaningful, we also compared the survival of immune subtypes from the TCGA database and found that the survival of immune subtypes (C1, C2, C3, C4, C6) in luminal A/B was not significantly different (*p* > 0.05) (Appendix A). From the results of the analysis of the survival of the subgroups and the clinical characteristics, it could be seen that different subgroups from the MSBR framework had different survival and degrees of malignancy.

### 2.4. The Immune Microenvironment in Subgroups

To explore the difference in immune environment, we calculated the proportion of infiltrating 22 immune cells in subgroups by using “cibersort”. In AIE, the proportion of CD8 T cells, NK cells, and M1 macrophages cells were higher than AEE, while the proportion of M2 macrophages cells and M0 macrophages cells was lower than AEE (*p* < 0.05); In luminal B, the proportion of BEE in M1 macrophages infiltration was lower and the proportion of BIE in CD8 T cells infiltration was higher, which was suspected to be related to the anti-tumor ability of “epithelial cell-enriched” was worse, and “immune cell-enriched” corresponded to T cell immunity in luminal BC (Figure 4A). We found that the expression of immune cell lysis activity (CYT) was significantly higher in the luminal IE subgroup, and AIE and BIE with high CYT showed high infiltration of anticancer immune cells, such as cytotoxic T cells, M1 macrophages and activated memory CD4+ T cells (Appendix A). In contrast, they had significantly lower infiltration of cancer-promoting immune cells, such as neutrophils and M2 macrophages, which was consistent with the results obtained in previous studies [20] (*p* < 0.05)

Furthermore, we mapped 1793 immune genes downloaded from the ImmPort database to 184/420 differential immune DNA methylation genes in luminal A/B and input these immune DNA methylation genes into “QDMR” (Appendix A). A total of 25/32 specific immune DNA methylation genes were identified and their DNA methylation levels showed significant differences in luminal A/B (*p* < 0.05). The levels of DNA methylation of immune genes between the two/three subgroups showed differences. In detail, DNA methylation levels of nearly all specific immune DNA methylation genes were higher in IE subgroups (AIE and BIE), whereas AEE, BEE and BNE showed lower DNA methylation levels in specific immune genes (Figure 4B,C).

Exploring the comparison of different subgroups at the point of immunosuppression (PD1/PDL1) is conducive to the selection of immunotherapy. The expressions of PD1/PDL1 in IE subgroups was higher than that in the other subgroups, and IE subgroups were more likely to benefit from immunotherapy (Figure 4D,E). Overall, we not only identified luminal subgroups, but also predicted subgroups for luminal A/B that could benefit from immunotherapy.

### 2.5. Differences in Pathological Characteristics of Subgroups

Considering significant differences in the immune microenvironment between subgroups may be manifested on pathology images, and we used “cellprofiler” software to perform feature extraction on processed pathological images in luminal A/B. A total of 1033 quantitative image features was selected for further analysis (Appendix A). The measurement standards of images in the subgroups were tested using the Kruskal–Wallis test. The luminal A/B subgroups showed significant differences in major image features. In detail, the values in the number of cells identified in AEE was higher than AIE (Figure 5A); BNE was higher in the “mean intensity” within the nucleus, BIE was higher in the “texture correlation,” BEE was higher in “eccentricity” (Figure 5B–D). TILs based on pathology images could reflect the distribution of lymphocytes and the intensity of the immune response. In details, the proportion of “non-brisk multifocal” type showed significant difference (*p* < 0.05) and was higher in AIE; the proportion of “brisk diffuse” type was relatively higher in BIE, and BNE was enriched in “brisk band-like” type, while the “non-brisk focal” type was the characteristic of BEE (Figure 5E,F; Appendix A). The results showed that differences at the molecular level between subgroups could also reflect differences in the spatial and morphological characteristics of their cells.

### 2.6. Subgroup Verification of GEO Data

The 22/11 “key genes” obtained from the MSBR framework were used to subgroup the external validation set from GSE72308 which included the 31/33 luminal A/B samples based on the hierarchical clustering method. We still obtained two and three subgroups respectively and these subgroups could be distinguished significantly (Figure 6A,B). To further validate the consistency in DNA methylation level of “key genes” and specific immune genes on the independent validation set, we compared the differences of DNA methylation levels of “key genes” and specific immune genes in subgroups. The results indicated that there were significant differences in the DNA methylation levels of 22/11 “key genes” (*p* < 0.05), which was consistent with the two/three subgroups in TCGA data (*p* > 0.05) (Figure 6C and Appendix A). In the DNA methylation level of most specific immune genes, AIE was significantly higher than AEE and subgroups in luminal B showed similar trends in TCGA and the independent validation set, indicating that luminal BC had broad DNA methylation and immune heterogeneity (*p* < 0.05) (Appendix A).

## 3. Discussion

BC heterogeneity increases the difficulty of diagnosis and treatment. Published studies have focused on the molecular subgroups of BC and have helped in the development of personalized treatments for specific new subtypes. For example, Zhang et al. used DNA methylation sites to identify nine prognostic subgroups by consensus clustering in BC [5]. Two novel immune subtypes were identified among TNBC patients [21]. However, these studies lacked extensive exploration of luminal BC with a strongly heterogeneous subtype. The development of a classification method for luminal BC is urgently needed for diagnosis and treatment.

The occurrence and development of cancer are accompanied by changes in DNA methylation patterns. Yizhak et al. showed that detection of gene methylation is more suitable as a marker for cancer screening [22]. In addition, research by Hinshelwood also shows the impact of epigenetics on BC [23]. In previous studies, DNA methylation has been used as a marker of tumor heterogeneity [24,25].

In our study, we used DNA methylation to develop an MSBR framework to accurately classify internal luminal A/B BC into different subgroups, and proved that there were significant differences in clinical stages, survival, immune heterogeneity and pathology features in these subgroups. Then, we used single-cell sequencing data to define the characteristics of luminal A/B subgroups and applied the deconvolution method based on gene expression data to predict AIE and BIE subgroups with a higher infiltration ratio of immune cells that were predicted to be more prone to immune benefit according to the comprehensive assessment of immune cell infiltration, PD-L1 expression, CYT and TILs. Biomarkers can be used to diagnose various disease. “Key genes” is important for diagnosing subgroups with immune characteristics, and “key genes” obtained by MSBR obtained consistent subgroups in independent validation sets, indicating that “key genes” are important in the diagnosis of precise subgroups. Specifically, AIE and BIE showed higher expression of PD-L1. Clinically, PD-L1 is currently the most widely used and accepted biomarker to guide the selection of patients to receive anti-PD-1 or anti-PD-L1 antibodies [26]. The results showed that IE subgroups may be prone to PD-L1 treatment. Moreover, CYT reflects the ability of cytotoxic T and NK cells to eliminate cancer cells and act as the new immunotherapy biomarker [27]. AIE and BIE with higher CYT have higher infiltration of anticancer immune cells, such as cytotoxic T cells, M1 macrophages and activated memory CD4+ T cells, and lower infiltration of cancer-promoting immune cells, such as neutrophils and M2 macrophages. Therefore, the immune characteristics in subgroups in our study provide a biological rationale for prioritizing checkpoint blockade-based therapy in patients with luminal BC. In addition, the heterogeneity of luminal BC may be more intuitively displayed on pathology images. We also quantified the characteristics of their pathology images and found significant differences in major features between subgroups, including texture and intensity. The quantitative features of these major pathologies have often been used to develop machine learning models for disease diagnosis [28]. BIE enriched the “brisk diffuse” TILs pattern, showing that it had a relatively strong immune infiltrate within the tumor. However, AIE enriched the “non-brisk multifocal” type, predicting AIE had a relatively weak immune response with loosely scattered TILs and this indicated its immune response may be weaker than luminal B. However, BEE enriched in “non-brisk focal” which reflected a very weak immune response. We also found that BNE was in an “intermediate state” in indicators of the immune microenvironment and was significantly enriched in “brisk band-like,” which may reflect its stronger immune performance, although not as effective as that of BIE. In conclusion, the present study showed that the MSBR framework could be applied to accurately subgroup luminal BC and demonstrated value that can be added by extensive analysis of molecular and quantitative imaging features and tumor lymphocytic infiltration patterns, providing directions for clinical diagnosis and immune treatment.

## 4. Materials and Methods

### 4.1. Data Source

We downloaded the level 3 DNA methylation data generated by the Illumina Infinium HumanMethylation450 BeadChip Array (HM450K) for BC patients from The Cancer Genome Atlas (TCGA GDC) database [29], and calculated Transcripts Per Kilobase Million values (TPM) for the mRNA expression profiles from the UCSC Xena database. The PAM50 subtype information and clinical information was downloaded from cBioPortal database [30]. We obtained 349/150 female BC luminal A/B samples, 343/542 non-luminal A/B samples based on the PAM50 subtype information and 96 normal samples for subsequent analysis. The non-luminal A/B are other BC subtypes with luminal A/B removed. Survival data used the standardized data set named TCGA Pan-Cancer Clinical Data Resource (TCGA-CDR) [31] recommended clinical outcome endpoint data for BC. In addition, GSE72308 [32] was used as an external validation dataset and GSE180286 single-cell sequencing data from The Gene Expression Omnibus (GEO) database. The 1793 immune genes were downloaded from the ImmPort database (https://www.immport.org/shared/home (accesed on 27 February 2018)).

### 4.2. Identify Differential DNA Methylation Sites and Genes

The genomic transcription initiation site upstream of 2000 bp and 500 bp downstream was used as the promoter region, and then we focused on gene DNA methylation data in the promoter region. The data was preprocessed to remove DNA methylation sites with missing beta values exceeding 50% of the total number of samples, and the K-Nearest Neighbor (KNN) method was used to supplement the null values. We calculated the mean DNA methylation levels β values of cancer samples and normal tissue samples respectively, and calculated the difference of the mean $\beta_{d}$. We also applied the Student’s *t*-test method to filter the differentially methylated sites between cancer samples and normal tissue samples. Finally, DNA methylation sites with $\left|\beta_{d}\right|\geq 0.2$ and false discovery rate (FDR)<0.05 were taken as the differential DNA methylation sites. The differential DNA methylation sites were then mapped to genes based on the HM450K platform annotation information.

### 4.3. Functional Annotation of Differential DNA Methylation Genes

We used the KOBAS online website (http://kobas.cbi.pku.edu.cn (accessed on 1 July 2011)) for The Kyoto Encyclopedia of Genes and Genomes (KEGG) pathway enrichment. KOBAS required us to input a set of gene symbols and returned the enriched KEGG pathways and *p* values. We selected the pathways with p value<0.05 as the significant functional pathways. The functional DNA methylation genes were included in the significant functional pathways.

### 4.4. The Construction of MSBR Framework for Subgrouping Luminal BC

First, 349/150 luminal A/B samples and 343/542 non-luminal A/B samples were randomly divided into 7:3 as two queues of training and test sets, respectively. We filtered the differential DNA methylation sites and removed the collinear DNA methylation sites from luminal A/B in the training set versus the normal samples.

We then set the random seed number, and applied the “Boruta” algorithm to obtain the feature DNA methylation sites for luminal A/B and normal samples by inputting the above differential DNA methylation sites. Considering that feature DNA methylation sites could more accurately represent the characteristics of luminal A/B, these feature DNA methylation sites were used to construct the SVM classifier for classifying luminal A/B and non-luminal A/B, and ten-fold cross-validation was applied for training the SVM classifier. Accordingly, the area under the curve (AUC) was used to evaluate the performance of the SVM classifier on the training, validation, and test sets. The feature DNA methylation sites were mapped to the DNA methylation genes and then were applied as feature DNA methylation genes based on the SVM classifier. Furthermore, to screen out key DNA methylation genes in luminal A/B associated with cancer progression, we integrated KEGG pathway enrichment analysis and weighted gene co-expression network analysis (WGCNA) [33]. We used differential DNA methylation genes between luminal A/B and normal samples to perform KEGG pathway analysis and WGCNA. Then, significant KEGG pathways and modules were obtained for luminal A/B. To select important cancer-related gene modules for analysis, we used the significantly enriched genes of the KEGG pathway related to cancer as background genes, and calculated the ratio of the number of genes in the module to the number of background genes as KMscore. KMscore represent an important degree in cancer progression. The higher the *KMscore*, the greater the importance of the module. Every module was further calculated $KM_{score_{i}}$ and key modules were selected for luminal A/B based on $KM_{score_{i}}$ from modules. The key modules were selected using the Algorithm 1.

The key modules obtained the high KMscore, and “key genes” were identified in the intersection of key modules and features in the SVM classification, which were considered to be the “key genes” of luminal A/B in BC, and these “key genes” could clearly distinguish the different subgroups in luminal A/B. The “NbClust” R package was used to determine the optimal cluster number [34]. Using the hierarchical clustering method, Luminal A/B were divided into different subgroups according to the DNA methylation level of the “key genes”, and the TSNE method was used for visualization. For more information about KMscore and “Boruta” feature selection, see Appendix A.

### 4.5. Single-Cell Signature Definition of Subgroups

To characterize the cellular microenvironment of subgroups, we used five cases of BC single-cell sequencing data from GSE180286 [19] to define characteristics of subgroups, and applied SingleR [35] to annotate cell type in BC. The “harmony” de-batching method was performed on the five cases of BC single-cell sequencing data, and the standardized single-cell sequencing data was clustered by “Seurat” R package [36] to obtain cell clusters. The standardized steps are implemented in the “NormalizeData” and “ScaleData” functions in Seurat. After the annotation of the “SingleR” software, the cell type of more than 50% of the cells in each cell cluster was used as the cell type. We used “FindAllMarkers” functions to obtain differential marker genes. The differential marker genes expression of each cell type was summed, resulting in a signature matrix. Next, we used a support vector machine-based deconvolution algorithm to input the gene expression profiles of the subgroups and the previously obtained signature matrix, and finally predicted the proportion of different cell types in BC, which were used to define subgroups characteristics in the BC cell microenvironment.

### 4.6. Compare the Clinical Stage Differences between the Subgroups

We extracted the clinical information of the corresponding subgroups, and calculated the proportion of each subgroup, showing the differences in the clinical stages between the subgroups in the bar plot. The chi-square test was used to analyze the subgroups in different stages—scope of the clinical primary tumor (T), presence or absence and scope of regional lymph node metastasis (N), and presence or absence and scope of regional lymph node metastasis (M). *p* value < 0.05 was applied as the threshold value to judge the significant difference.

### 4.7. Survival Analysis between Subgroups

Using the recommended event endpoint progression-free survival (PFS) in BC, we performed the survival analysis between the subgroups in luminal A/B, and used the Kaplan–Meier (KM) method to analyze the survival differences in subgroups in luminal A/B. *p* value < 0.05 was used as the threshold to compare the significance of survival differences. The immune subtype information was obtained from a previously study [37].

### 4.8. Analysis of Immune Gene Methylation Levels and Immune Cell Infiltration between Subgroups

A total of 1793 immune genes were downloaded from the ImmPort database (https://www.immport.org/home (accessed on 27 February 2018)) and mapped to 420 differentially methylated immune genes. We used Quantitative Differentially Methylated Regions (QDMRs) to obtain a set of immune gene lists with different methylation levels between different subgroups, and performed “cibersort” by using gene expression data to obtain the immune infiltration ratio of 22 kinds of cells for comparison.

### 4.9. Calculation of Immune Cell Lysis Activity (CYT)

CYT is the mean value of the gene expressions of the GZMA and PRF1.

### 4.10. Features Extraction of Pathology Images

The pathology images of 349/150 luminal A/B samples were downloaded and were converted to PNG format. Then, we used “cellprofiler” software [38] to measure relevant image features in luminal A/B subgroups by inputting pathology images. The images were automatically segmented using an ‘IdentifyPrimaryObjects’ module and an ‘IdentifySecondaryObjects’ module to identify the cell nuclei and cytoplasm. Six modules were applied to obtain quantitative features, including “IdentifyPrimaryObjects,” “IdentifySecondaryObjects,” “Measure ObjectSizeShape,” “MeasureTexture,” “MeasureObjectIntensity,” and “Measure ObjectIntensityDistribution”. Finally, “cellprofiler” software outputs relevant feature measurements in cell size and shape, cell texture and pixel intensity of the cell. In addition, we compared TILs based on H&E staining images between the luminal A/B subgroups according to Saltz et al. [39]. TILs can reflect the distribution of lymphocytes and intensity of immune response and it were defined as 4 types, which were “brisk band-like,” “brisk diffuse,” “non-brisk focal,” “non-brisk multifocal”.

### 4.11. Statistical Analysis

*T*-test method to filter differentially methylated sites in cancer and normal tissue samples. We defined differentially methylated sites or genes with *p* < 0.05 and FDR < 0.05. The chi-square test was implemented in R software to test the differences in the TILs type between subgroups (*p* < 0.05), and the Kruskal–Wallis test was used to compare differences between subgroups.

## Figures and Tables

**Figure 1 ijms-23-12747-f001:**
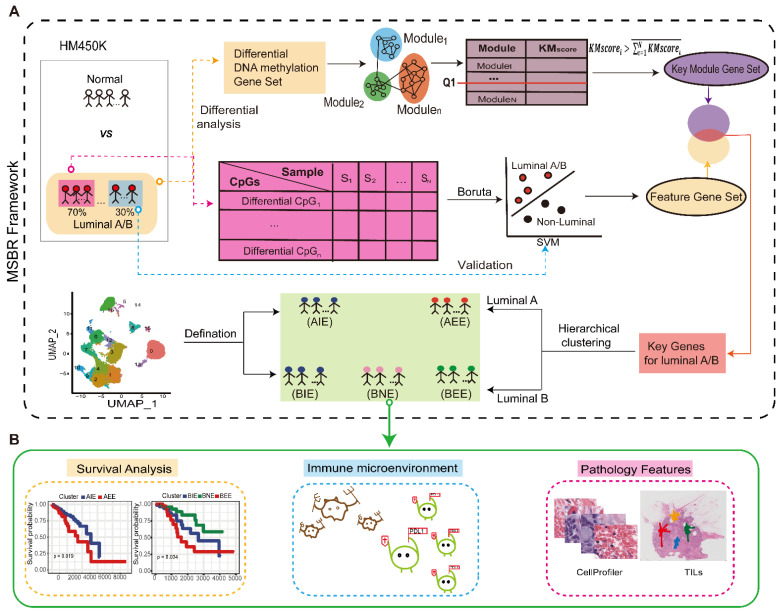
Overview of the Study. (**A**) Workflow of MSBR framework based on key module scoring algorithm and “Boruta” method. (**B**). Analysis of breast cancer subgroups in survival, immune microenvironment, and pathological image features.

**Figure 2 ijms-23-12747-f002:**
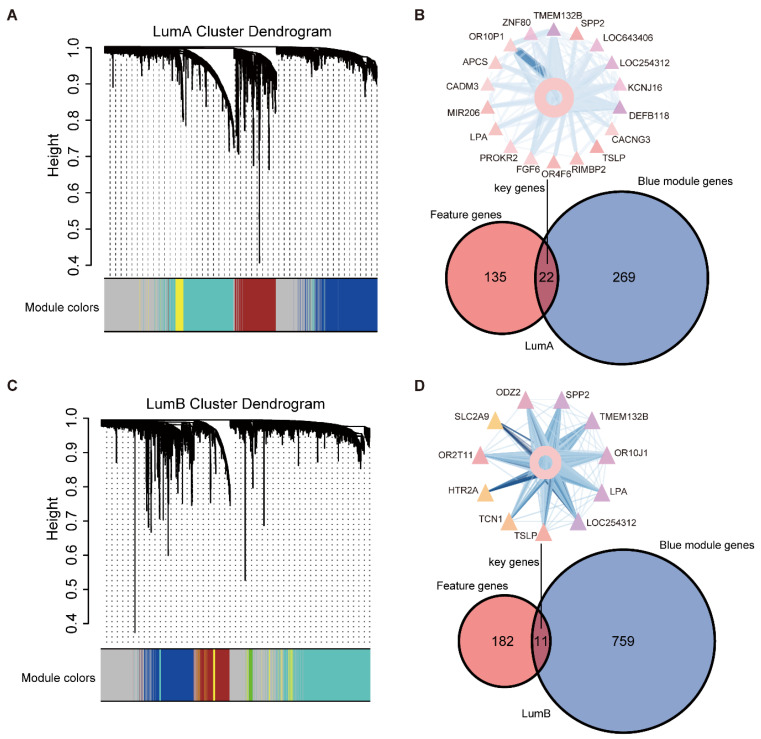
Acquisition of “key gene”. (**A**) Differential DNA methylation module in luminal A. (**B**) 22 “key genes” for subgrouping luminal A. (**C**) Differential DNA methylation module in luminal B. (**D**) 11 “key genes” for subgrouping luminal B.

**Figure 3 ijms-23-12747-f003:**
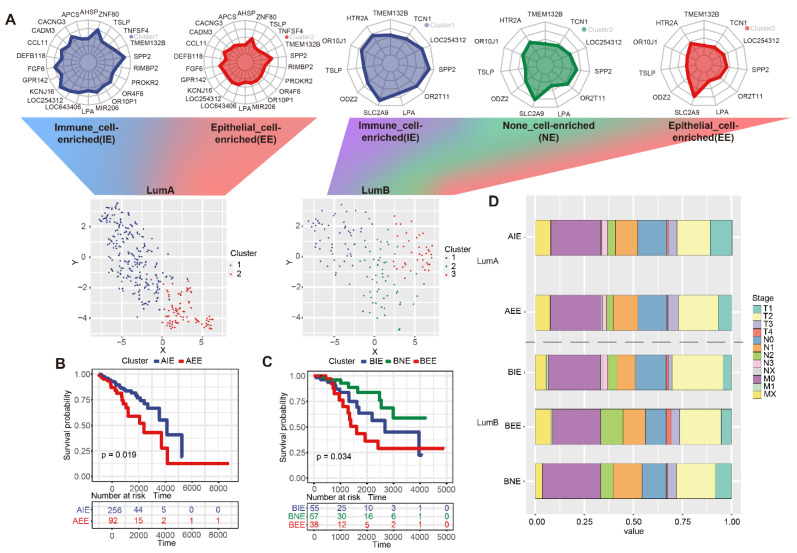
The definition of subgroups in luminal A/B. (**A**) Subgroups of luminal A/B. (**B**) Survival difference of two subgroups in luminal A. (**C**) Survival difference of three subgroups in luminal B. (**D**) Comparison of clinical stages between subgroups in luminal A/B.

**Figure 4 ijms-23-12747-f004:**
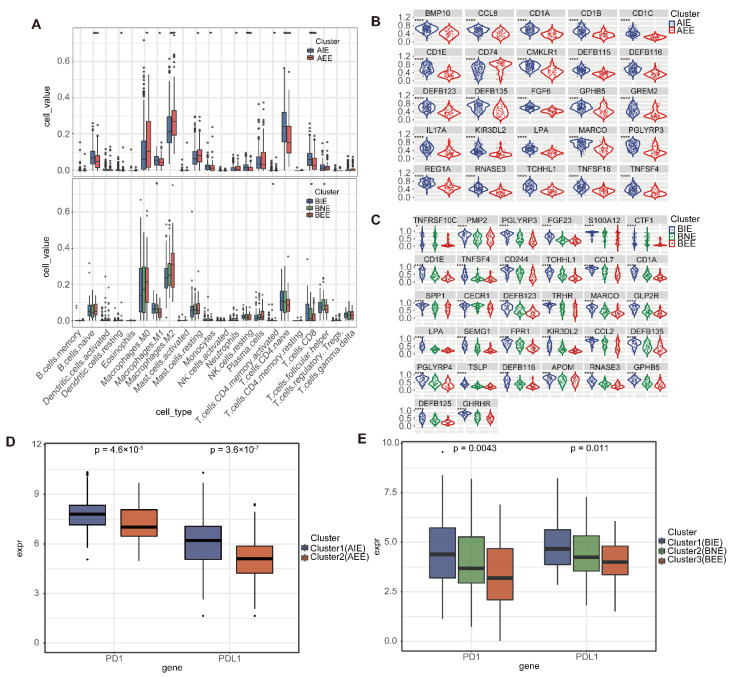
The immune microenvironment in luminal A/B subgroups. (**A**) Comparison of 22 kinds of immune cells infiltration ratio between subgroups. (**B**) Comparison of DNA methylation of specific immune genes between luminal A subgroups. (**C**) Comparison of DNA methylation of specific immune genes between luminal B subgroups. (**D**) The expression of PD1/PDL1 of subgroups in luminal A. (**E**) The expression of PD1/PDL1 of subgroups in luminal B. (* *p* < 0.05. *** *p* < 0.001. **** *p* < 0.0001.).

**Figure 5 ijms-23-12747-f005:**
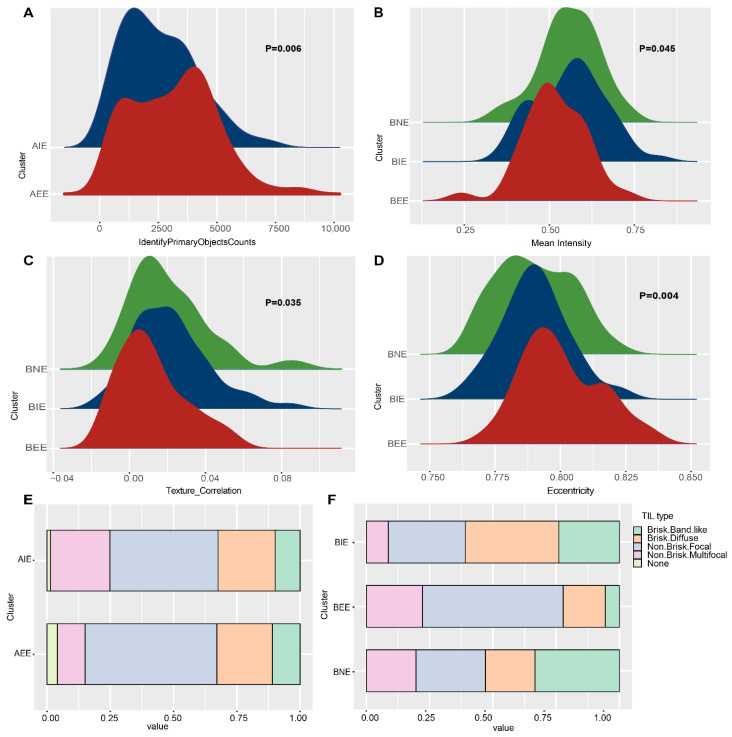
Images differences in luminal A/B subgroups. (**A**) The differences in “IdentityPrimaryObjectsCounts” between luminal A subgroups. (**B**) The differences in “Mean Intensity” between luminal B subgroups. (**C**) The differences in “Texture Correlation” between luminal B subgroups. (**D**) The differences in “Eccentricity” between luminal B subgroups. (**E**) The differences of TILs type between luminal A subgroups. (**F**) The differences of TILs type between luminal B subgroups.

**Figure 6 ijms-23-12747-f006:**
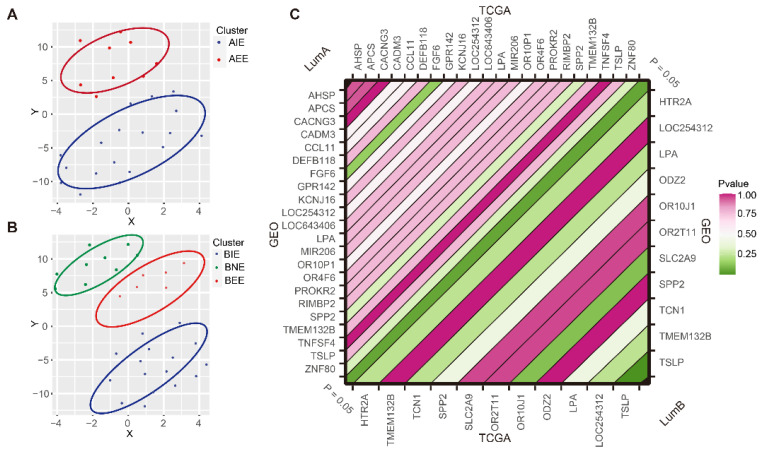
The subgroups in GEO dataset and TCGA data. (**A**) TSNE showed luminal A subgroups in GEO. (**B**) TSNE showed luminal B subgroups in GEO. (**C**) The concordance of distribution on DNA methylation gene levels in TCGA and GEO dataset.

## Data Availability

The source code and data generated in this study has been deposited in github [https://github.com/HIT-CBC/BSNet.git (accessed on 2 September 2022)], The code is implemented by Python and R.

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
