# Peer review of "Classification of Subgroups with Immune Characteristics Based on DNA Methylation in Luminal Breast Cancer"

_ijms, 2022, doi:10.3390/ijms232112747_

Round 1

Reviewer 1 Report

The paper of Zhang et al. is an interesting classification based on epigenetics and immune characteristics of luminal breast cancer. The paper is well written, the introduction is detailed and the analysis well performed.

Below, are minor revisions for the authors:

Why was split the BC Luminal A into 2 subgroups (AIE/AEE), using the computational framework MSBR, while luminal B was split into 3 (BIE/BEE and None-cell enriched, BNE)? Elucidate the role of BNE in the analysis.

Fig. 3. I suggest using the same colors for the graphic representation of the clusters of Lum A and Lum B shown in Fig. 3A and 3B/C. For example, blue for IE and NE, red for EE, and green for NE. In this way, it is easier to compare the same cluster from different luminal BC.

Figure S5. The results showed that different subgroups have different survival and degrees of malignancy. Which type cluster? Based on? C1 who represents? Please, insert a description in the figure legend or in the main text.

Finally, the discussion section is not totally convincing and needs a more detailed focus on the result obtained, highlighting the role provided by this study on directions for clinical diagnosis and immune treatment.

Author Response

Dear professor:

Best regards.

Reviewer 2 Report

In the present manuscript, the authors identified the key module gene set of differentially methylated genes in luminalA/B according to their importance in pathway and co-expression; and then obtain the feature gene set that is most critical for clustering. The two are taken to intersect to obtain the key genes, and the key genes were used to divide luminalA/B into new subgroups, and then various methods are used to verify the difference and rationality of the divided subgroups.

They observed that the subgroups had significant differences in DNA methylation levels, immune microenvironment (immune cell infiltration, immune checkpoint PD1/PDL1 expression, immune cell cracking activity and pathology features.

Main points:

1. When selecting the feature gene, the "boruta" algorithm is used to select the feature for classifying luminal A/B and non-luminal A/B classifier of the feature. However, the key of the "boruta" algorithm, the dependent variable used in the selection, is not mentioned directly, so we can only speculate that it should be consistent with the classification of luminal or non-luminal in validation. Moreover, it is said in the paper that the training set and test set are 7:3 luminal A/B samples, so where do the non-luminal samples in training and testing come from?

2.  The selection of Module gene may change significantly with the adjustment of KMscore requirement range, because the selection method of gene set does not seem to be rigorous and thorough, and it should be adjustable; the selection of feature gene may also be affected by the sample (the process of "remove collinear site" process may also affect?) The intersection of the two is very small. The intersection of the two is only a very small number: lumA:22,lumB:11; and this small number of key gene is unstable with the collection selection, as long as there are one or two changes, the situation of the final subgroup may be obviously different; and the article does not seem to have a large number of different experiments to prove the stability of the method. Therefore, the article feels more like "finding a division that just works" rather than "discovering a method that can produce reasonable and effective divisions".

Minor points:

1, Possible text error, all other places are 349/150 corresponding to luminal A/B, only line 98 has no A.

2, Line 347 "the number of set Gene_i" may be misspelled, maybe it should be "the number of elements of set Gene_i";

In line 352, {KMscore_i,KMscore_2,...... , KMscore_N}, why the subscript of the first item is i instead of 1, (there should be nothing special about gene_1 or KMscore_1 that needs to be excluded from the calculation of the mean).

If the subscript is 1, is that consistent with {KMscore_i,i=1,... ,N} in line 353; and if so, whether to consider not using two different representations to cause confusion.

For the selection of the key module, the text seems to lack a description of its design and principle, only saying that "the key module got high KMscore" (whether the score is used to measure what should be explained in the text); when selecting the gene set to be added to the KMgene, the KMscore is less than or equal to the up quartile. The KMscore is less than or equal to the up quartile limit setting, which is not even in line with the default inference that "key module" corresponds to "high score", so it should not be ignored.

Author Response

Dear Professor:

Best regards.

Round 2

Reviewer 2 Report

The authors have addressed all my comments. In the revised version they have  addressed all major points  as far as it concerns me.